# Exploring New Tools for Risk Classification among Adults with Several Degrees of Obesity

**DOI:** 10.3390/ijerph20136263

**Published:** 2023-06-30

**Authors:** Greice Westphal-Nardo, Jean-Philippe Chaput, César Faúndez-Casanova, Carlos Alexandre Molena Fernandes, Eliane Cristina de Andrade Gonçalves, Raquel Tomiazzi Utrila, Karine Oltramari, Felipe Merchan Ferraz Grizzo, Nelson Nardo-Junior

**Affiliations:** 1Department of Physical Education, Associate Graduate Program in Physical Education UEM/UEL, Health Sciences Center, State University of Maringa, Maringa 87020-900, Parana, Brazil; cfaundez@ucm.cl (C.F.-C.); nnjunior@uem.br (N.N.-J.); 2Healthy Active Living and Obesity Research Group, Children’s Hospital of Eastern Ontario Research Institute, Ottawa, ON K1H 8L1, Canada; jpchaput@cheo.on.ca; 3Center for Multiprofessional Studies on Obesity—NEMO/HUM/UEM, University Hospital of Maringa, State University of Maringa, Maringa 87083-240, Parana, Brazil; carlosmolena126@gmail.com (C.A.M.F.); elianeandradegoncalves@gmail.com (E.C.d.A.G.); raqueltomiazzi@hotmail.com (R.T.U.); karineoltramari@gmail.com (K.O.); fmfgrizzo@uem.br (F.M.F.G.); 4Faculty of Education Sciences, Catholic University of Maule, Talca 34809112, Maule, Chile

**Keywords:** obesity, risk assessment, metabolic syndrome, AIP, HOMA-IR, risk stratification

## Abstract

The epidemic of obesity worldwide has been recognized as a very important challenge. Within its complexity, the identification of higher-risk patients is essential, as it is unsustainable to offer access to treatment to all people with obesity. Several new approaches have recently been presented as important tools for risk stratification. In this research, we applied several of these tools in a cross-sectional study involving adults with obesity classes I, II, III, and super-obesity. The participants had their cardiometabolic risk profiles assessed. The study included adults with obesity aged 18 to 50 years (*n* = 404), who were evaluated using anthropometric, body composition, hemodynamic, physical fitness, and biochemical assessments. These variables were used to identify the prevalence of risk factors for cardiometabolic diseases according to the classes of obesity by gender and age group. The results showed high prevalence of risk factors, especially among the upper classes of obesity (BMI > 35 kg/m^2^) using single parameters as the waist circumference, with almost 90% above the cut-off point. For smaller numbers such as Glycated Hemoglobin, however, the prevalence was around 30%. Indexes such as the atherogenic index of plasma (AIP) had the highest prevalence, with 100% of the male participants identified as being at increased risk for cardiovascular disease.

## 1. Introduction

The highly complex etiology of obesity and its dynamic encompassing genetic, physiologic, environmental, psychological, social, economic, and even political factors interact in several ways to promote and aggravate the obesity epidemic [1,2,3]. For this reason, it is hard to treat obesity effectively [4]. It is well known that excess adipose tissue, particularly ectopic fat deposits, are implicated in more than 200 complications of obesity and negatively impact the health of affected individuals [5].

The high and rising prevalence of obesity, along with its associated health impacts, represents a real challenge for health and policy authorities, as it can be economically prohibitive to offer treatment access to all people in need [6,7,8]. Additionally, obesity is not necessarily synonymous with health risks, as the metabolically healthy obese (MHO) phenotype and the fat-but-fit paradigm have been documented [9,10].

In fact, there is consistent evidence about the prevalence of MHO, with certain studies presenting very high rates of that phenotype, such as one with Brazilian women in which the prevalence of MHO was around 70% when considering the HOMA-IR and NCEP-ATPIII criteria for metabolic syndrome [6]. The prevalence of this phenotype can vary between ≈15% and ≈30% depending on the definition of Metabolic Syndrome (MetS) relied upon, using the criterion of meeting either 0 or 0–1 MetS components [10]. Studies have revealed that MHO is an unstable state, as an important portion of MHO subjects evolved to an unhealthy phenotype within a number of years [9,11]. The necessity of finding better ways to diagnose people with obesity through the use of risk stratification categories is important in determining which people are at high risk and in need of intervention strategies. The traditional criteria used to diagnose MetS require the presence of three or more of five components: greater waist circumference (WC), dyslipidemia with high triglycerides (TG), low high-density lipoprotein cholesterol (HDL-C) levels, elevated blood pressure (BP), and impaired fasting plasma glucose (FPG) [12]. Because of its dichotomous or binary nature, authors have proposed an option in which a continuous metabolic syndrome risk score can be used. This approach has the advantage of preserving the statistical power, which is decreased when dichotomizing continuous variables. In addition, it allows more precise measurement of the MetS risk along a continuum [12,13,14].

The one-size-fits-all approach should be avoided when programs to treat obesity are offered. The identification of subgroups with distinct risk profiles is an important way to improve clinical practice [7,10]. It is essential to make it more feasible to offer access to treatment programs to those with higher health risks [3]. Hence, the main objective of this study is to assess the prevalence of traditional and recently developed risk factors by assessing tools such as the triglyceride glucose (TyG) index and related indexes, the continuous metabolic severity score (MetSs), and the atherogenic index of plasma (AIP), which are related to different obesity categories, in a sample of Brazilian adults.

## 2. Materials and Methods

This descriptive study was a cross-sectional design carried out with 404 adults with obesity of both sexes aged between 18 and 50 years (Body Mass Index—BMI ≥ 30 kg/m^2^). Participants were selected to take part in the research project during the years 2018, 2019, 2020 (first semester), and 2022 (second semester) through the Multidisciplinary Treatment of Obesity Program (MTOP) coordinated by the Multidisciplinary Obesity Studies Nucleus (NEMO) of the State University of Maringá (UEM) and Regional University Hospital of Maringá (HUM). A detailed description of this study can be found elsewhere [15]. In brief, eligible participants were invited to take part voluntarily in the study through dissemination in the local media (TV, radio, newspaper) and social networks (website and institutional email, Facebook). The interested participants took part in a pre-inclusion phase (Cardiometabolic Risk Assessment; CAR, divided into two steps) to confirm their eligibility for involvement in the study. In step 1, the conditions of eligibility were verified (age over 18 and under 50 and BMI over 30 kg/m^2^; furthermore, it was required that they exhibit abdominal obesity, specifically, a waist circumference exceeding 88 cm for women and 102 cm for men). In all, 774 people answered an anamnesis which included socioeconomic and health data. The evaluation consisted of the following: body mass, height, BMI, waist circumference (WC), and body composition by bioimpedance. In addition, their blood pressure (BP) and basal heart rate (HR) were measured. Finally, health-related physical fitness tests, including the sit and reach for flexibility, the 30 s sit and stand for lower limbs resistance, the plank strength test for abdominal static resistance, and the six-minute walking test (6MW) for cardiorespiratory fitness, were applied.

After this process, a total of 404 people met the inclusion criteria and were considered eligible to participate in step 2, which included carrying out laboratory tests to verify their cardiometabolic risk profiles through the fasting measures of blood glucose, insulinemia, glycated hemoglobin, total cholesterol, HDL-c, LDL-c, VLDL-c, triglycerides, and ultrasensitive C-reactive protein.

In order to determine the dosages of these biochemistry variables, standard procedures were applied by specialized professionals from a private laboratory with quality control and an ISO certification.

Beyond the single parameters mentioned above, other surrogate measures of Insulin Resistance (IR) were determined by the Homeostasis Model Assessment (HOMA-IR), calculated as follows: HOMA-IR = (insulin × glucose)/22.5 [16,17]. The evaluation of homeostasis to verify the beta cells of the pancreas was determined by the calculation (Homa-Beta): 20 × Insulin (iu/mL) ÷ (Glycemia − 3.5), and the cut-off reference values were between 167 to 175 [18].

The triglyceride glucose (TyG) index was calculated as ln [fasting triglycerides (mg/dL) × fasting plasma glucose (mg/dL)/2]) [19]. We calculated the product of triglyceride (TG) and fasting plasma glucose (FPG), the TyG index, the TyG related to the adiposity status obtained by the equation (TyG/body mass index), and the TyG related to visceral adiposity by the ratio of TyG to waist circumference [20]. In addition, TyG was used for the risk assessment involving the atherogenic index of plasma (AIP), which is defined as the logarithm of plasma triglycerides to HDL-c ratio [21].

For calculation of the continuous metabolic syndrome severity z scores (MetSs) the following information was entered for each participant: birthdate, sex, race/ethnicity, height, weight, waist circumference, systolic blood pressure, HDL-c, triglycerides, and fasting glucose, using the calculator provided by Gurka et al. (2017) at the website (http://mets.health-outcomes-policy.ufl.edu/calculator/, accessed on 2 March 2023 [22].

All procedures followed the requirements of Resolution 466/2012 of the National Health Council for research involving human beings, which is based on the principles of the Helsinki Declaration. All participants read and signed the Term of Free and Informed Consent agreeing to voluntarily participate in the research. The research was previously approved by the Permanent Committee of Ethics in Research of the State University of Maringá (Record no. 2,655,268).

The researchers involved in the assessments were all trained and followed standard procedures for measuring anthropometric variables with proper tools, such as height with a wall stadiometer (Sanny^®^, Canastota, NY, USA), waist circumference (WC) with a flexible anthropometric tape (Medical Starrett-SN-4010 model, Sanny^®^), and body weight with a bioimpedance electric device (InBody^®^, model 520 Body Composition Analyzers, Seoul, Republic of Korea). Blood pressure was measured using an automatic arm monitor (model HEM-7113, Omron^®^, Kyoto, Japan). Blood collection and analysis for measuring blood glucose, high-density lipoprotein (HDL-c), and triglycerides was performed by qualified professionals in a private clinical analysis laboratory with quality certification between 7:00–9:00 a.m. and with patients observing fasting for at least 8 h.

To calculate the BMI, we used the following formula: weight (kg)/[height (m) × height (m)]; classification was based on the cutoff points of the World Health Organization (WHO, 2011). The WC measurement rating was based on the WHO cutoff points as well, with WC > 94 cm for men and >80 cm for women indicating increased risk of metabolic complications and WC > 102 cm for men and >88 cm for women indicating substantially increased risk of metabolic complications [23].

Blood pressure was classified according to the Seventh Brazilian Guideline on Hypertension, specifically, normotension: systolic blood pressure (SBP) and diastolic blood pressure (DBP) ≤ 120/80 mmHg; prehypertension: SBP between 121 and 139 and/or DBP between 81 and 89 mmHg; and hypertension: SBP ≥ 140 mmHg and/or DBP ≥ 90 mmHg [24].

Classification of fasting blood glucose followed the criteria of the Guidelines of the Brazilian Society of Diabetes 2017–2018, specifically, normoglycemia: fasting blood glucose < 100 mg/dL; pre-diabetes (or increased risk for diabetes mellitus): ≥100 to <126 mg/dL; and established diabetes: ≥126 mg/dL (Oliveira, Montenegro Junior and Vencio 2017). The lipid profile was classified according to the 2017 Brazilian Guideline for Dyslipidemia and Atherosclerosis Prevention, specifically, high level of fasting triglycerides: ≥150 (mg/dL) and low fasting HDL-c level: <40 mg/dL for men and <50 mg/dL for women [25].

For data analysis, normality was verified using the Kolgomorov–Smirnov test. The average (χ) and standard deviation (SD) were used as descriptive statistics. To compare variables according to gender, unpaired *t*-tests were used. To compare the variables according to age group and level of obesity, a one-way ANOVA was used with Bonferroni correction for multiple comparations, while post hoc tests were used to indicate those groups between which there were differences. To correlate the variables, the Pearson correlation coefficient was used. Analyses were performed using the Statistical Package for Social Sciences (SPSS)^®^ version 20.0 [26]. A significance level of *p* < 0.05 was adopted for all analyses.

## 3. Results

The participants in this study were 404 adults with obesity (Table 1), 85 men (21%) and 319 women (79%) aged between 18 and 50 years (mean ± SD: 36.6 ± 8.8 years). The BMI of the sample ranged from 31.3 to 77.2 kg/m^2^ (mean ± SD: 42.5 ± 6.7). According to the WHO cutoff points, 76 (18.8%) had grade 1 obesity (BMI > 30.0 < 35 kg/m^2^), 141 (34.9%) had grade 2 obesity (BMI > 35.0 < 40 kg/m^2^), 160 (39.6%) had grade 3 obesity (BMI > 40.0 < 45 kg/m^2^), and 27 (6.7%) had grade IV or super obesity (BMI > 45 kg/m^2^). Considering the degree of obesity, significant differences (*p* < 0.05) were observed in all anthropometric/body composition variables except for height and the relationship of lean mass to fat mass. At the same time, the classes and degree of obesity presented significant differences related to the hemodynamic and physical fitness variables, with SBP being higher in the group with obesity class IV and heart rate (HR) being higher in the group with obesity class III compared to class I (*p* < 0.01). There were significant differences in the total distance on the six minute walking test, with the group with obesity class IV presenting the lowest distance (*p* < 0.01). Flexibility was lower in the groups with obesity class III and class IV compared to those with class I and class II (*p* < 0.01).

The biochemistry parameters indicated significant differences in glucose, insulin, HOMA-IR, Homa-beta, and hs-PCR among the participants with obesity class III and class IV compared to those with class I and class II, with the higher BMI classes presenting the less healthy results (*p* < 0.01). However, neither the markers of dyslipidemia nor the HbA1c levels showed significant differences. Lastly, related to the index or ratios applied to identify alterations related to insulin resistance (IR) or dyslipidemia, significant differences were observed in the percentiles of the continuous metabolic syndrome scores related to BMI, with the higher classes of obesity presenting the more severe scores with regard to risk of metabolic syndrome (*p* < 0.01). The same pattern was observed in the MetS-WC and Percentile of MetS-WC. The TYG parameters have a similar pattern as well, with the groups with obesity classes III and IV presenting higher risk (*p* < 0.01).

When looking at the data stratified by age group (Table 2), which were organized in two categories consisting of young adults (age below 40 years old) and middle-aged (age over 40 years), the differences between these groups are evident and significant (*p* < 0.05), with higher values for the younger group in height, body mass, BMI, lean body mass, absolute body fat, and waist, abdomen, and hip circumferences. Among the hemodynamic and physical fitness variables, a significant difference was only observed for systolic blood pressure, with a higher average for the middle-aged group (*p* < 0.01). Concerning the biochemical parameters, significant differences were observed for glycemia, Homa β, total cholesterol, HDL-c, LDL-c, Non-HDL Cholesterol, and Glycated Hemoglobin, all of which had higher values for the middle-aged group (*p* < 0.01). The only exception was for Homa β, for which the middle-aged group presented a lower average. Finally, for the indices derived from biochemical/anthropometric parameters significative differences were observed only for TYG and TYG-BMI.

Table 3 and Figure 1 and Figure 2 show the proportion of participants presenting alterations in single parameters and biochemical indexes. Lower prevalence was observed for HbA1c, with 34.1% of men and 31.3% of women showing elevated values (above of the cutoff point of 5.7%) for that parameter [27]. On the other hand, the prevalence was higher for the AIP index, with 100% of men and 92.8% of women presenting values above the moderate risk cutoff point of 0.1–0.24 and >0.24 at high risk of cardiovascular disease [28].

It can be noted in Figure 3 that the severity score of metabolic syndrome (MetSs) and the atherogenic index of plasma (AIP) presented a significant correlation, with a coefficient of determination (R^2^) of 69%, indicating that the two indexes are strongly related.

## 4. Discussion

The main goal of this study was to verify the risk profile of participants in one multidisciplinary treatment program for obesity (MTPO) offered regularly by the Maringa State University to different age groups since 2005. The adults involved in that program had real need of professional care to treat their obesity and comorbidities. In Maringa, there is no public funded program offering that kind of treatment to this public. Therefore, this population needs to volunteer as research subjects in order to have access to this model of assistance. This is the first and maybe the most important data to be presented from this program. The reality is not very different from that observed in most of the countries around the world, which typically do not have MTPOs to assist their population [8].

The second important information provided by this study is the high prevalence of metabolic risk found in this sample, as 100% of the male participants were classified as having moderate or high risk of cardiovascular disease according to the AIP. The atherogenic lipoprotein profile of plasma has been recognized as a substantial risk factor for atherosclerosis due to the significance of triglycerides in atherosclerotic and cardiovascular disease. In addition to individual serum cholesterol levels, the atherogenic index of plasma (AIP) has been suggested as a marker of plasma atherogenicity based on the evidence of its positive association with lipoprotein particle size, cholesterol esterification rates, and remnant lipoproteinemia [30].

The AIP has been found to be one of the strongest markers in predicting cardiovascular disease (CVD) risk. There is evidence indicating that AIP is associated with other CVD risk factors as well [31]. Thus, MTPO to promote lifestyle modification is strongly recommended, and the monitoring of parameters such as AIP can be used to assess the efficacy or effectiveness of this kind of intervention program [31].

It is worth explaining that the AIP is defined as the logarithm of the ratio of triglycerides to high-density lipoprotein cholesterol (HDL-C). Thus, it is a strong predictor of future cardiovascular disease. AIP is directly and independently associated with arterial stiffness, and is known to be inversely correlated with LDL particle size. In addition, it can be readily calculated from routine lipid profiles [21].

There is great interest in the development of new and comprehensive lipid indexes, such as the atherogenic index of plasma, which might reflect the balance between atherogenic and anti-atherogenic factors. Recently, AIP has been shown to be a strong marker for predicting the risk of CAD, with the value of AIP being positively associated with waist circumference and BMI and inversely associated with physical activity [32].

Another recent study found evidence to propose that AIP can be considered as a novel and better biomarker for obesity, as subjects in the higher quartiles of AIP all had a significantly increased risk of obesity compared with those in the lowest quartile in a Chinese population [33]. Interestingly, the mean value for the group with obesity in that study was 0.13, whereas in our study the same mean was 0.44, more than three times higher. We identified a similar fact when comparing the results from our study to another applying the AIP in a Chinese population with coronary artery disease, in which the average AIP was 0.17, compared to 0.12 in the control group [32].

Therefore, results such as these are very important in balancing the idea of metabolic healthy obesity (MHO), which can mislead and postpone the intervention programs that a high number of individuals need. Even when the MHO phenotype is correctly identified, there is a chance of its reversing over time, as MHO is an unstable state and an important part of the subjects evolve to an unhealthy phenotype within a number of years [9,11].

It is important to highlight that even simple measures such as waist circumference can be very useful as a clinical tool; for instance, this single parameter diagnosed more than 90% in this study regardless of gender. A single inexpensive variable, alone or combined with more complex models, can be very useful in risk stratification. It is important to express, however, that while increased WC is undeniably an excellent marker of cardiometabolic status in individuals with normal weight or overweight, it is less useful in individuals with higher BMI [3].

In this context, it becomes relevant to remind the reader that studies have revealed that abdominally obese males increase their risk of cardiovascular disease twentyfold over the course of five years, reinforcing the necessity of protocols including simple measures such as WC as important tools to identify high risk individuals [7,34]. The same authors have demonstrated that excess fat occurs predominantly due to subcutaneous or visceral abdominal fat, which are closely related to insulin resistance (IR) and diabetes.

Another important finding of this study is a positive association between the degree of obesity and the resting heart rate (HR). Additionally, our results revealed that individuals in higher BMI categories exhibited lower performance on the six-minute walking test (6MWT). These findings emphasize the crucial role of multi-component interventions in facilitating improvements in these variables and mitigating cardiometabolic risks [35].

The data presented in our study corroborate the prevalence of HOMA-IR above the cutoff point, at over 80% in both genders, with men presenting the higher prevalence (89.4%) and similar numbers for women (86.2%). This is an important number, as insulin resistance is one of the main factors associated with cardiovascular disease. It is additionally important because in Brazil this diagnosis test is not included in routine exams, and as such an important proportion of persons with pre-diabetes or diabetes are not being identified at the proper time [36].

It is important to obtain information about the utility of HOMA-Beta, as it has been proposed as a measure of the functionality of pancreatic beta cells. It has a cutoff point of 167 to 175, with lower values interpreted as indicating better beta cell functioning. Based on this cutoff point, it was shown that a very high proportion of the study population was exposed to higher risks of long-term damage to beta cells functioning, and consequently to higher risks of developing DM2 [18].

Another single parameter which showed very high prevalence was hsCRP, with numbers close to 90% above the cutoff point of 3.0 mg/L, which is considered a high-risk category for cardiovascular diseases [25]. Such high levels reinforce the proinflammatory state promoted by the excess of adipose tissue, which is common among people with obesity. Our data made clear a strong relationship between the degree of obesity (BMI classes) and the proinflammatory state, with the average hsCRP being higher as BMI class increases, as shown in Table 1. These data reinforce the need to monitor people with obesity, as hsCRP has been recognized as the most important biomarker associated with cardiovascular risk [37]. In a systematic review, hsCRP was used to stratify risk in several chronic non-communicable diseases, resulting in the conclusion that hsCRP is the most promising of the seric biomarkers in these conditions and can be employed to assess both the clinical status and evolution. This can be especially valuable information in programs seeking to promote health for people with obesity, both in risk stratification and as a biomarker related to the effectiveness of those programs [38].

Despite the overall importance of the traditional methods of diagnosing metabolic syndrome (MetS), there are difficulties related to the utility of this information for present or future (follow-up) use due to its binary nature. In recognition of this, other ways to use the information have been presented, such as the continuous metabolic syndrome risk score (cMSy), which may be a more appropriate and valid alternative for epidemiological and clinical studies [12,13,39]. This score can be interpreted as a Z-score (mean 0, SD = 1), with higher scores corresponding to a higher risk of MetS [14,39]. Authors revealed racial/ethnic discrepancies using the traditional MetS criteria, finding that certain racial groups with diabetes have a low prevalence of MetS and high MetS severity scores, which is a contradictory result. The same authors found that the MetS severity score was correlated with risk of future type 2 diabetes and CVD [40]. More important perhaps is the fact that their study showed no meaningful differences between the MetS-Z-WC and MetS-Z-BMI scores in terms of their association with future CHD and T2DM, indicating the potential utility of MetS-Z-BMI for clinical use.

This kind of tool has great potential for monitoring prospectively high-risk patients, as it reveals that compared to individuals with a change in score smaller than (<0), participants with a change greater than half a point (>0.5) had a hazard ratio (HR) of 2.66 for incident diabetes (*p* < 0.001). Our data show an increasing mean of both MetS-Z-WC and MetS-Z-BMI scores among the BMI classes, with the higher BMI classes presenting significantly higher averages (Table 1), especially related to MetS-Z-WC. Considering this, individuals with more severe MetSs can be directed to more intense MPTO and it can easily be determined whether the program has effectively improved their condition simply by the reduction in the score, which is much easier to explain than the traditional dichotomic model.

This is in line with the fact that the American Diabetes Association and the European Association for the Study of Diabetes have recommended using the continuous MetS risk score for investigating the association of MetS with potential risk factors in children and adolescents [41].

Considering the above, Gurka et al. (2017) have developed equations to calculate a standardized metabolic syndrome severity score for different populations. These equations load coefficients for the five metabolic syndrome components and transform them into a single metabolic syndrome factor that can be used to generate a score for each subgroup. They have made available a calculator for this mean which can be accessed by the following website: http://mets.health-outcomes-policy.ufl.edu/calculator/, accessed on 25 March 2023 [22].

The metabolic syndrome severity scores are z scores (normally distributed and ranging from theoretical negative to positive infinity with mean = 0 and SD = 1). Authors have demonstrated that these scores are strongly correlated with other markers of risk, such as hsCRP, uric acid, and the HOMA-IR. Moreover, it has been confirmed that they are correlated with long-term risk of CVD and diabetes [14].

The data comparing younger adults with the middle-aged group shows that despite the younger group having a higher average BMI, the cardiometabolic risks are more relevant in the middle-aged group. These results are aligned with those of other studies, in which the age has been found to have an important role in increasing risk [42,43,44]. This should be considered in MPTO and in policies addressing this issue.

Overall, our results reinforce the higher risk of adults with obesity of contracting chronic non-communicable diseases, which has frequently been neglected, postponing essential actions to avoid these risks and shortening lifespan, as has been demonstrated by longitudinal studies such as that of Greenberg et al. (2013), which found an early mortality of 9.4 years among people with obesity compared to the normal weight group [45]. The same pattern was shown by Kitahara et al. (2014), whose data demonstrated that early mortality is strongly associated with the degree of obesity, with an average lifespan shorter by 6.5 years for the group with BMI between 40 to 44.9 kg/m^2^, 8.9 years for the group with BMI between 45 and 49.9, and around 13.7 years for the group with BMI over 55 kg/m^2^ [46]. The same authors showed that different criteria, even simple ones such as WC, can be very important tools for assessment, and they can have their diagnostic ability improved by combining the results with more sophisticated parameters such as the ratios and indexes presented in the present study.

Finally, it is important to recognize that this study has limitations, notably its cross-sectional design, which does not allow conclusions about causality to be drawn; therefore, our findings should be confirmed by follow-up studies. In addition, there are imbalances in the gender of the participants and the BMI classes, especially with respect to the group with obesity class IV. An additional limitation is the absence of adequate control over participants’ medication usage. On the other hand, this made it possible to verify that the study population can face very challenging situations, reflected by the high risk profile they presented, that may have been ignored due to the fact that obesity has been recognized as a very neglected disease around the world [8].

## 5. Conclusions

This study brings forward important information about the risks associated with obesity and the relevance of the degree of obesity in a relatively young group from a developing country. It makes clear that more attention is needed to ensure that evaluation and risk stratification in this group align with policies to promote multidisciplinary treatment programs for obesity funded by public agencies. In addition, it presents values that can be used as references and for comparison with different populations along with a variety of traditional and recently presented parameters for clinical and epidemiological purposes. Finally, it highlights how impacted the health of these people can be, and how necessary it is to bring about changes in the way that they are treated.

## Figures and Tables

**Figure 1 ijerph-20-06263-f001:**
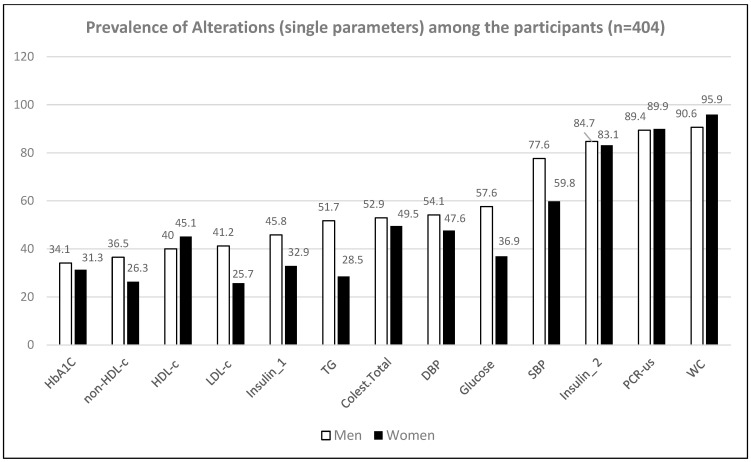
Prevalence of alterations in single parameters used in the diagnosis of metabolic risk.

**Figure 2 ijerph-20-06263-f002:**
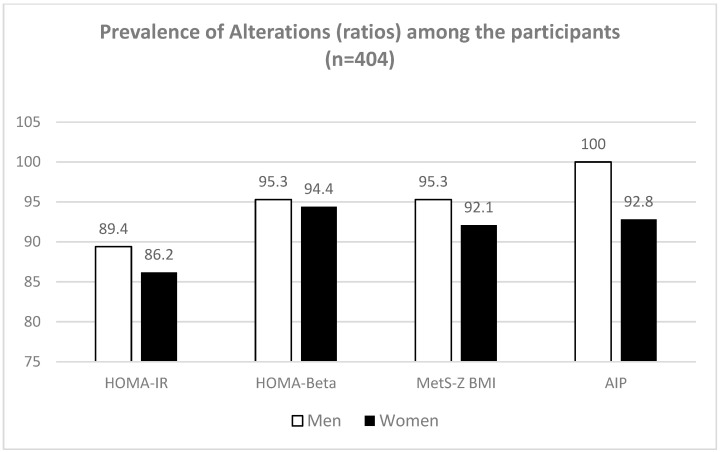
Prevalence of alterations in ratios or indexes based on the combination of single parameters used in the diagnosis of metabolic risk.

**Figure 3 ijerph-20-06263-f003:**
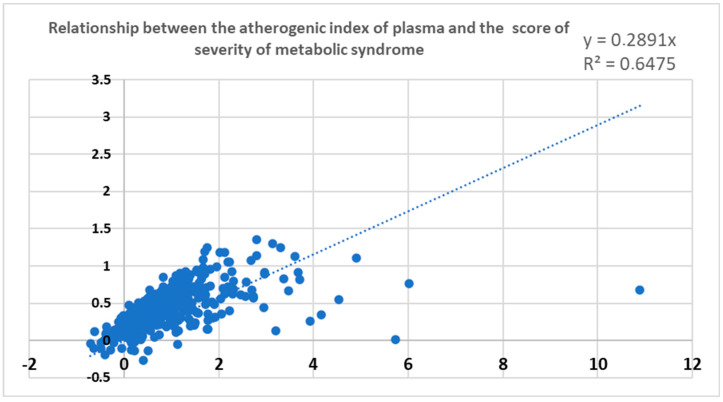
Relationship between two indexes of metabolic risk, the Atherogenic Index of Plasma (AIP) and the Metabolic Syndrome Severity Score (MetSs).

**Table 1 ijerph-20-06263-t001:** Cardiometabolic risk variables according to obesity classes among the studied sample of Brazilian adults (*n* = 404).

Antrometric and Body Composition Parameters	Obesity Class I (*n* = 76)	Obesity Class II (*n* = 141)	Obesity Class III (*n* = 160)	Super Obesity (*n* = 27)	*p*-Value
Age (years)	41.66 ± 8.75	39.50 ± 9.10	38.27 ± 8.77	37.15 ± 8.32	0.028 *
Height (m)	162.92 ± 8.03	163.97 ± 8.69	165.04 ± 8.11	165.85 ± 10.64	0.23
Body Mass (kg)	88.38 ± 9.86 ^a.b.c^	100.20 ± 11.27 ^a.d.e^	119.14 ± 15.53 ^b.d.f^	156.03 ± 25.99 ^c.e.f^	<0.001 *
Body Mass Index (kg/m^2^)	33.16 ± 1.28 ^a.b.c^	37.32 ± 1.48 ^a.d.e^	43.70 ± 2.65 ^b.d.f^	56.88 ± 6.60 ^c.e.f^	<0.001 *
Lean Body Mass (kg)	37.21 ± 13.96 ^b.c^	39.29 ± 15.82 ^d.e^	47.20 ± 2.65 ^b.d.f^	59.29 ± 22.31 ^c.e.f^	<0.001 *
Skeletal Muscle Mass (kg)	29.91 ± 6.98 ^b.c^	31.21 ± 6.14 ^e^	32.93 ± 5.82 ^b.f^	39.18 ± 6.99 ^c.e.f^	<0.001 *
Body fat (%)	45.62 ± 5.28 ^b^	48.60 ± 5.17	53.86 ± 31.70 ^b^	55.67 ± 2.54	0.009 *
Absolute Body Fat (kg)	51.17 ± 12.98 ^a.b.c^	60.91 ± 13.31 ^a.d.e^	71.93 ± 15.97 ^b.d.f.^	96.74 ± 19.28 ^c.e.f^	<0.001 *
Lean Mass/Body Fat Ratio (kg)	0.82 ± 0.43	0.71 ± 0.37	0.71 ± 0.29	0.64 ± 0.24	0.052
Neck Circumference (cm)	37.91 ± 3.91 ^b.c^	39.01 ± 4.06 ^e^	40.73 ± 4.33 ^b.f^	43.46 ± 4.92 ^c.e.f^	<0.001 *
Waist Circumference (cm)	96.27 ± 7.56 ^a.b.c^	103.86 ± 9.78 ^a.d.e^	114.24 ± 10.05 ^b.d.f^	130.67 ± 16.72 ^c.e.f^	<0.001 *
Abdomen Circumference (cm)	105.40 ± 7.62 ^a.b.c^	114.62 ± 10.52 ^a.d.e^	126.81 ± 10.83 ^b.d.f^	147.63 ± 15.32 ^c.e.f^	<0.001 *
Hip Circumference (cm)	117.03 ± 10.56 ^a.b.c^	124.37 ± 12.75 ^a.d.e^	131.80 ± 11.06 ^b.d.f^	148.41 ± 16.21 ^c.e.f^	<0.001 *
Waist/Height Ratio (cm)	0.59 ± 0.04 ^a.b.c^	0.63 ± 0.05 ^a.d.e^	0.69 ± 0.05 ^b.d.f^	0.79 ± 0.09 ^c.e.f^	<0.001 *
**Hemodynamic/Health Related Physical Fitness Variables**					
Systolic Blood Pressure (mmHg)	123.25 ± 14.70 ^c^	127.32 ± 14.52	127.41 ± 13.78	132.25 ± 11.79 ^c^	**0.027 ***
Diastolic Blood Pressure (mmHg)	79.67 ± 12.08	81.65 ± 12.54	82.43 ± 10.27	84.85 ± 12.49	0.178
SPO_2_ (%)	96.71 ± 3.36	96.51 ± 12.54	96.04 ± 2.35	92.26 ± 1.77	0.197
HR (bpm)	77.80 ± 8.89 ^b^	80.76 ± 12.87	84.06 ± 11.26 ^b^	84.89 ± 12.66	**0.001 ***
Six Minutes’ Walk Test (m)	505.33 ± 86.25 ^c^	496.02 ± 73.95 ^e^	485.63 ± 70.27 ^f^	431.83 ± 81.54 ^c.e.f^	**<0.001 ***
Plank Strength Test (s)	28.88 ± 26.85	27.96 ± 24.54	25.31 ± 22.81	17.05 ± 14.55	0.116
Dynamic Lower Limb Muscular Endurance (n rep.)	15.72 ± 4.54	15.16 ± 4.69	14.40 ± 3.78	13.78 ± 4.29	0.064
Flexibility (cm)	22.79 ± 8.14 ^b.c^	19.62 ± 9.86 ^d^	14.67 ± 7.77 ^b.d^	15.14 ± 10.27 ^c^	**<0.001 ***
**Biochemical Parameters**					
Glycemia (mg/dL)	95.25 ± 12.16 ^b^	101.73 ± 30.87	111.96 ± 50.52 ^b^	106.70 ± 31.43	**0.010 ***
Insulin (mU/L)	18.68 ± 9.15 ^c^	23.02 ± 11.39	22.35 ± 10.99	28.52 ± 14.89 ^c^	**0.001 ***
Homa IR	4.45 ± 2.46 ^b.c^	5.73 ± 3.13	6.22 ± 4.42 ^b^	7.25 ± 3.60 ^c^	**0.001 ***
Homa β	67.34 ± 32.87 ^c^	81.68 ± 44.17	74.49 ± 38.71 ^f^	99.86 ± 58.88 ^c.f^	**0.002 ***
US-CRP (mg/L)	4.02 ± 3.44 ^b.c^	5.81 ± 5.35	7.52 ± 6.50 ^b^	8.45 ± 5.43 ^c^	**<0.001 ***
Total cholesterol (mg/dL)	192.74 ± 40.01	190 ± 36.17	196.22 ± 38.30	179.78 ± 38.65	0.161
HDL-c (mg/dL)	49.92 ± 12.25	46.73 ± 11.99	46.74 ± 12.36	48.78 ± 15.78	0.236
LDL-c (mg/dL)	117.08 ± 36.94	113.84 ± 30.74	119.05 ± 31.51	107.47 ± 30.33	0.259
VLDL-c (mg/dL)	23.99 ± 11.34	27.88 ± 15.19	28.68 ± 15.79	23.16 ± 8.41	0.051
Non-HDL Cholesterol (mg/dL)	140.34 ± 39.93	141.78 ± 36.03	149.40 ± 36.76	135.93 ± 37.63	0.12
Triglycerides (mg/dL)	127.53 ± 65.03	145.55 ± 84.86	158.25 ± 106.17	126.15 ± 74.53	0.061
Glycated Hemoglobin (%)	5.52 ± 0.54	5.66 ± 0.98	5.79 ± 1.44	5.50 ± 0.79	0.288
**Indices Derived From Biochemical/Anthropometric Parameters**					
AIP (mg/dL)	0.37 ± 0.26	0.45 ± 0.27	0.47 ± 0.29	0.38 ± 0.28	0.067
MetS-Z BMI	0.38 ± 0.55	0.86 ± 0.79	1.35 ± 1.24	1.67 ± 0.66	0.059
Percentile BMI	63.35 ± 18.91 ^a.b.c^	75.34 ± 15.43 ^a.d.e^	83.70 ± 13.35 ^b.d.f^	92.52 ± 6.84 ^c.e.f^	**<0.001 ***
MetS-Z WC	0.17 ± 0.60 ^a.b.c^	0.60 ± 0.83 ^a.d^	1.03 ± 1.17 ^b.d^	1.10 ± 0.67 ^c^	**<0.001 ***
Percentile WC	55.43 ± 21.08 ^a.b.c^	67.39 ± 18.57 ^a.d.e^	75.83 ± 17.73 ^b.d^	82.30 ± 13.05 ^c.e^	**<0.001 ***
TYG (mg/dL)	8.59 ± 0.53 ^b^	8.74 ± 0.58	8.87 ± 0.71 ^b^	8.65 ± 0.57	**0.011 ***
TYG-BMI	284.92 ± 21.22 ^a.b.c^	326.44 ± 25.10 ^a.d.e^	387.97 ± 42.01 ^b.d.f^	493.35 ± 75.64 ^c.e.f^	**<0.001 ***
TYG-WC	828.17 ± 94.33 ^a.b.c^	909.97 ± 117.98 ^a.d.e^	1015.51 ± 137.87 ^b.d.f^	1133.68 ± 179.47 ^c.e.f^	**<0.001 ***

^a^ Obesity I vs. Obesity II; ^b^ Obesity I vs. Obesity III; ^c^ Obesity I vs. Super Obesity; ^d^ Obesity II vs. Obesity III; ^e^ Obesity II vs. Super Obesity; ^f^ Obesity III vs. Super Obesity; One-Way Anova test and Bonferroni post hoc were used for differences; SPO^2−^ oxygen saturation; HR-heart rate; bpm: beats per minute; mg/dL—milligrams per deciliter; us-CRP—ultra-sensitive C-reactive protein; HDL—high density lipoprotein; LDL—low density lipoprotein; AIP—atherogenic index of plasma; BMI—Body Mass Index; TYG—Triglyceride-Glucose Index; WC—Waist Circumference. Data are shown as mean values ± standard deviations. * *p*-value < 0.05.

**Table 2 ijerph-20-06263-t002:** Cardiometabolic risk variables stratified by age group among the studied sample of Brazilian adults (*n* = 404).

Antrometric and Body Composition Parameters	Young Adults (*n* = 197)	Middle Age Adults (*n* = 207)	*p*-Value
Age (years)	31.84 ± 5.26	46.32 ± 5.17	**<0.001 ***
Height (m)	166.16 ± 7.36	162.57 ± 9.12	**<0.001 ***
Body Mass (kg)	114.90 ± 22.65	103.78 ± 20.57	**<0.001 ***
Body Mass Index (kg/m^2^)	41.61 ± 6.98	39.19 ± 5.79	**<0.001 ***
Lean Body Mass (kg)	45.19 ± 17.70	41.63 ± 17.20	**0.041 ***
Skeletal Muscle Mass (kg)	32.68 ± 6.30	31.69 ± 6.84	0.131
Body fat (%)	52.38 ± 28.83	48.89 ± 5.57	0.088
Absolute Body Fat (kg)	69.71 ± 20.22	62.15 ± 16.06	**<0.001 ***
Lean Mass/Body Fat Ratio (kg)	0.71 ± 0.33	0.73 ± 0.36	0.613
Neck Circumference (cm)	39.88 ± 4.34	39.42 ± 4.46	0.296
Waist Circumference (cm)	109.79 ± 13.46	106.94 ± 13.42	**0.034 ***
Abdomen Circumference (cm)	122.42 ± 14.55	117.53 ± 15.44	**0.001 ***
Hip Circumference (cm)	129.92 ± 13.87	125.92 ± 14.45	**0.019 ***
Waist/Height Ratio (cm)	0.66 ± 0.07	0.65 ± 0.07	0.679
**Hemodynamic/Health Related Physical Fitness Variables**			
Systolic Blood Pressure (mmHg)	124.46 ± 12.41	129.26 ± 15.43	**0.001 ***
Diastolic Blood Pressure (mmHg)	80.89 ± 10.36	82.65 ± 12.66	0.128
SPO_2_ (%)	96.55 ± 2.13	96.15 ± 2.75	0.102
HR (bpm)	82.87 ± 12.36	80.75 ± 11.41	0.075
Six Minutes’ Walk Test (m)	495.51 ± 77.21	483.51 ± 76.81	0.118
Plank Strength Test (s)	24.91 ± 21.61	27.72 ± 25.84	0.237
Dynamic Lower Limb Muscular Endurance (n rep.)	14.61 ± 4.16	15.12 ± 4.45	0.239
Flexibility (cm)	18.46 ± 8.82	17.47 ± 9.77	0.289
**Biochemical Parameters**			
Glycemia (mg/dL)	99.86 ± 27.56	109.67 ± 45.89	**0.010 ***
Insulin (mU/L)	23.31 ± 11.03	21.34 ± 11.50	0.081
Homa IR	5.79 ± 3.38	5.77 ± 3.97	0.966
Homa β	88.22 ± 41.80	71.75 ± 41.33	**0.006 ***
US-CRP (mg/L)	6.58 ± 6.1	6.08 ± 5.33	0.376
Total cholesterol (mg/dL)	184.65 ± 35.05	199.56 ± 39.37	**<0.001 ***
HDL-c (mg/dL)	45.53 ± 12.22	49.31 ± 12.47	**0.002 ***
LDL-c (mg/dL)	110.99 ± 29.17	120.93 ± 34.33	**0.002 ***
VLDL-c (mg/dL)	26.20 ± 14.03	28.04 ± 14.96	0.202
Non-HDL Cholesterol (mg/dL)	138.40 ± 34.11	149.58 ± 39.43	**0.003 ***
Triglycerides (mg/dL)	143.02 ± 98.22	148.62 ± 83.27	0.536
Glycated Hemoglobin (%)	5.40 ± 0.74	5.92 ± 1.34	**<0.001 ***
**Indices Derived From Biochemical/Anthropometric Parameters**			
AIP (mg/dL)	3.58 ± 3.28	3.34 ± 2.48	0.41
MetS-Z BMI	0.97 ± 0.86	1.06 ± 1.17	0.267
Percentile BMI	77.34 ± 17.53	77.73 ± 16.71	0.818
MetS-Z WC	0.65 ± 0.82	0.78 ± 1.12	0.168
Percentile WC	68.82 ± 19.99	70.10 ± 20.19	0.524
TYG (mg/dL)	4.68 ± 0.32	4.76 ± 0.31	**0.012 ***
TYG-BMI	361.83 ± 70.31	346.85 ± 59.86	**0.021 ***
TYG-WC	955.34 ± 154.02	947.52 ± 152.18	0.608

*t* test for independent samples; SPO2—oxygen saturation; HR-heart rate; bpm: beats per minute; mg/dl—milligrams per deciliter; us-CRP—ultra-sensitive C-reactive protein; HDL—high density lipoprotein; LDL—low density lipoprotein; AIP—atherogenic index of plasma; BMI—Body Mass Index; TYG—Triglyceride-Glucose Index; WC—Waist Circumference. Data are shown as mean values ± standard deviations. * *p*-value < 0.05.

**Table 3 ijerph-20-06263-t003:** Proportion of participants presenting alteration in single parameters and biochemical indexes.

Single Parameter	Men (*n* = 85)	Women (*n* = 319)
Glycated Hemoglobin (%)	34.1	31.3
Non-HDL Cholesterol (mg/dL)	36.5	26.3
HDL-c (mg/dL)	40	45.1
LDL-c (mg/dL)	41.2	25.7
Insulin (mU/L) *	45.8	32.9
Triglycerides (mg/dL)	51.7	28.5
Total cholesterol (mg/dL)	52.9	49.5
Diastolic Blood Pressure (mmHg)	54.1	47.6
Glycemia (mg/dL)	57.6	36.9
Systolic Blood Pressure (mmHg)	77.6	59.8
Insulin (mU/L) **	84.7	83.1
US-CRP (mg/L)	89.4	89.9
Waist Circumference (cm)	90.6	95.9
**Index or ratios**	**Man (*n* = 85)**	**Women (*n* = 319)**
Homa IR	89.4	86.2
Homa β	95.3	94.4
MetS-Z BMI	95.3	92.1
AIP (mg/dL)	100	92.8

HDL—high density lipoprotein; LDL—low density lipoprotein; Insulin (mU/L) *—Cut-off point 23.0 Loria et al. (2010) [29]; us-CRP—ultra-sensitive C-reactive; Insulin (mU/L) **—Cut-off point 12.2 Mcauley et al., 2001; AIP—atherogenic index of plasma; BMI—Body Mass Index. Data are presented as proportion (%).

## Data Availability

Not applicable.

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
