# Peer review of "Exploring New Tools for Risk Classification among Adults with Several Degrees of Obesity"

_ijerph, 2023, doi:10.3390/ijerph20136263_

Round 1
Reviewer 1 Report
I carefully read and reviewed the paper titled "Exploring New Tools to Risk Classification Among Adults with Several Degrees of Obesity", (ijerph-2402425). Authors performed a quality study in obese subjects to find out cardiovascular risks in this population. My comments are as follows:
1. Title, abstraction of the text and keywords look fine.
2. Introduction is a bit long but includes all relevant information related to the background of the study.
3. Methodology was expressed very well. Statistics are correct.
4. Results were ease to follow. However, I suggest expressing the p values of 0.000 as <0.001 in the tables. Moreover, there are two table 1 in the text. I think the latter one should be numbered as table 2.
5. Authors discussed their results in line with important aspects of the topic. However, I recommend them discussing inflammation's role since it is both associated with obesity (Elife. 2023 May 5;12:e83069. doi: 10.7554/eLife.83069), and cardiovascular outcomes (Postepy Kardiol Interwencyjnej. 2018;14(3):263-269. doi: 10.5114/aic.2018.78329). Could inflammation have an effect on the increased cardiovascular risk of obese subjects?
6. Limitations should be moved to the last paragraph of discussion. Otherwise, conclusions are justified.
Reviewer 2 Report
Dear authors,
Westphal-Nardo et al. explore the utility of different classical physical and metabolic markers in stratifying cardiovascular risk in people with different degrees of obesity.
Even when the study is sound, I have some concerns regarding the novelty of this data and how some of the results are interpreted or the absence of consideration of some other interesting data presented here. Please find my comments and suggestions below:
Page 2, line 59: please amend “sore” for “score”.
Considering that most of the outcomes measured here are sensitive to change by the action of pharmacology, this information should be declared in this study to understand the behavior of the data.
Figures 1 and 2 are not necessary since the same data is summarised in Table 3.
The calculation of the score of severity of metabolic syndrome (MetSs) is not properly explained in the methods section. Please add it for the reproducibility of the study.
In the figure 3, please add the p value of the prediction equation.
Lines 247 to 250: please rewrite this sentence since the word “correlation” is quite repetitive.
It is not clear the rationale behind the comparisons between young and middle-aged adults and between men and women. Please explain it.
Line 270-271: Which is the data or reference that support this statement?: “The AIP was found to be one of the strongest markers in predicting the cardiovascular disease (CVD) risk”.
Line 275-277: This information was already given in the text: “It is worth to explain that the AIP is defined as the logarithm of triglycerides to high-density lipoprotein cholesterol (HDL-C) ratio. Thus, it is a strong predictor of future cardiovascular disease”.
Please amend the title of figure 3: Atherogenic Index Of Plasma atherogenic Index, it should be atherogenic index of plasma
It is not clear how can be proposed that AIP is a good marker to stratify cardiovascular risk in obesity if all the participants in your sample were positive. Is it possible that a good marker for this purpose should be one that increases at every obesity level?
Overall, the results presented here are not different from what has been classically presented previously, where at a higher level of obesity, the metabolic disorders worsen. However, one of the novel results presented here is that physical fitness measurements don’t change or worsen at higher levels of obesity, and that was not even mentioned in the discussion section. Please add your analysis regarding this topic.
No limitations were addressed in the text, (please add them in the last paragraph of the discussion, not in the conclusion), where a couple of examples are the inequality in terms of the gender of the participants and the lower number of subjects in the obesity level 4.
Please review the format of the reference list, since some of them have a different format.
The quality of English is sufficient and it doesn't require additional revisions.
Reviewer 3 Report
It is interesting work, although most of the findings have already been described. The inclusion of novel risk markers provides interesting information. However, the achievement of some objectives is not clear and the presentation of the results must be improved since they are incomplete. Some observations:
Introduction.
One of the main objectives was to assess the continuous metabolic severity scores (MetSs), which is not clear in the document.
Methods.
Selection criteria. ¿What were the inclusion criteria? Please write them down; authors only mention that 404 subjects met the inclusion criteria (line 90).
Results.
Tables 1 and 2.
In the column of "p" values there is an asterisk (*) in the significant values, but authors do not mention what it refers to.
In the column Hemodynamic variables the location of the Diastolic and Systolic pressure are incorrect; they are in place of each other.
Please explain in the text the meaning and how you get the MetS-ZBMI and MetS-WC score. ¿Are these the variables that refer to the continuous metabolic severity scores?
The BMI and WC percentiles are not continuous variables. It is better to use the median and interquartile range.
Table 3.
Please specify the criteria of Egidio et al. and Mcauley et al. for insulin values. The word "al" is missing in Mcauley (et al).
Figure 1.
Please specify to which values the white and black bars correspond.
Figure 2.
¿What are series 1 and 2 of the frequencies shown in the figure?
Figure 3.
¿ What is the axis that corresponds to the values of the score of severity of metabolic syndrome and and which to that of the atherogenic index of plasma?
¿How did you obtain the score of severity of metabolic syndrome?
No comments.
Round 2
Reviewer 2 Report
Dear authors,
thank you for taking into consideration my comments and suggestions. I don't have any further concerns with this manuscript.
Author Response
Dear reviewer, thank you for your invaluable feedback. It has greatly improved the clarity and comprehensiveness of our work, enhancing its academic rigor and integrity. Your guidance has been instrumental in refining our research.
Reviewer 3 Report
Thank you very much for the responses to my comments. The manuscript is now clearer.
However, the inclusion criteria are still not clear to me, or if you can write the paragraph better: you indicate that in step 1 the conditions of eligibility were “age over 18 and under 50 and BMI over 30 kg/m2” and 774 subjects were evaluated for anthropometry, body composition, blood pressure and the health related physical fitness tests. Then 404 subjects “met the inclusion criteria”. It is not clear, then, what the inclusion criteria were for Step 2 ¿What was the reason why the 774 subjects did not continue in the study? ¿Were they subjects who did not complete the aforementioned evaluations?
Regarding the variables (BMI and WC) that are expressed in percentiles, I did not comment that they did not have a normal distribution, but that they are not continuous variables, so it is better to express their distribution by reporting the median and interquartile range.
Author Response
Dear reviewer, thank you for your invaluable feedback. It has greatly improved the clarity and comprehensiveness of our work, enhancing its academic rigor and integrity. Your guidance has been instrumental in refining our research.
Regarding the inclusion criteria, your question is very timely. In addition to age and BMI, it was also necessary to include abdominal obesity, meaning a waist circumference greater than 88cm for women and greater than 102cm for men. That was explained in the text, Line 85-87.
Once again, I express my gratitude for your attention and suggestions. Your contribution has been crucial in addressing important aspects of the study.